# Protective and Therapeutic Effects of Orlistat in Combination with *Elettaria cardamomum* “Cardamom” Extract on Learning, Memory, Anxiety, and Neuroinflammation in Obese Mice

**DOI:** 10.3390/medicina61020263

**Published:** 2025-02-04

**Authors:** Anfal AL-Dalaeen, Nour Batarseh, Nadine N. Abdelhadi, Sally Atawneh, Reem AbuKashef, Ali Mosa Ra-shid Al-Yasari

**Affiliations:** 1Department of Clinical Nutrition and Dietetics, Faculty of Allied Medical Sciences, Applied Science Private University, Amman 11937, Jordan; 2Department of Clinical Nutrition and Dietetics, Faculty of Allied Medical Sciences, Philadelphia University, Amman 19392, Jordan; nbatarseh@philadelphia.edu.jo; 3Department of Clinical Pharmacy and Pharmacy Practice, Faculty of Pharmacy, Aqaba University of Technology, Aqaba 771111, Jordan; nabdulhadi@aut.edu.jo; 4Department of Clinical Pharmacy and Therapeutics, Applied Science Private University, Amman 11937, Jordan; sally.atawneh.sa@gmail.com; 5Faculty of Medicine, University of Jordan, Amman 11942, Jordan; reem.abukashef@gmail.com; 6College of Veterinary Medicine, Al-Muthanna University, Samawah 66001, Iraq; alialyasari@mu.e.du.iq

**Keywords:** learn, memory, orlistat, neuroinflammation, elettaria cardamomum

## Abstract

*Introduction and Objective*: Obesity has increased worldwide, and existing anti-obesity medications have treatment limitations that diminish their overall benefits. This study aimed to investigate the effects of orlistat in combination with *Elettaria cardamomum* “Cardamom” (CAR) extract on working memory, recognition memory, anxiety, and inflammation within hippocampal tissue. *Methods*: Mice were categorized into two groups: a control group (CD) and a cafeteria diet (CAF) group induced with obesity (CAF) for 10 weeks. The groups were then subdivided into a CAF group treated with orlistat (CAF-ORL), a CAF group treated with orlistat and *Elettaria cardamomum* (CAF-ORL-CARD), and a group that continued on the CAF. The CAF-ORL group received orlistat at a dosage of 10 mg/kg/day for four weeks, while the CAF-ORL-CARD group received 10 mg/kg/day of orlistat and 500 mg/kg of CAR extract via oral gavage. In the 14th week, various assessments were conducted, including the novel object recognition (NOR) test, Y maze test, marble-burying test (MBT), open-field test, and TNF-α levels in the hippocampus. *Result*: TNF-α levels in the hippocampal tissue of the CAF group were elevated compared to the CD group (*p* < 0.01), whereas the CAF-ORL group exhibited reduced TNF-α levels compared to the CAF group (*p* < 0.01). Moreover, TNF-α levels in the CAF-ORL-CARD group were significantly lower than in the CAF-ORL group (*p* < 0.01). The recognition index was notably higher in the CAF-ORL group compared to the CAF group (*p* < 0.01) and higher in the CAF-ORL-CARD group compared to the CAF-ORL group (*p* < 0.01). However, there were no changes in the open-field test and Y maze test (*p* > 0.05). *Conclusions*: Orlistat combined with CAR has positive effects on neuroinflammation and memory, suggesting that this combination may offer potential therapeutic benefits for cognitive impairments and hippocampal dysfunction associated with obesity.

## 1. Introduction

Obesity has become a major epidemic and a worldwide health issue. The prevalence of obesity among adults is estimated to exceed 40% and is expected to increase further in the aftermath of the COVID-19 pandemic [1]. Research shows that obesity raises the risk of neurodegenerative diseases by contributing to the deterioration of neural membrane glycerophospholipids, disrupting protein synthesis and degradation, and increasing the production of reactive oxygen species and brain lip toxicity [2,3,4].

The administration of orlistat demonstrated that it modifies the levels of various hypothalamic neuropeptides within the central nervous system (CNS) [5] or alters the concentrations of essential appetite-regulating monoamine neurotransmitters, thereby leading to appetite suppression [6]. Additionally, treatment with a herbal mixture contributed to the protection of the brain from damage, attributed to its omega-3 fatty acid content [7].

Spices are widely recognized for their ability to enhance gastric function by stimulating salivary production and the secretion of gastric juices, thereby aiding in the digestive process. *Elettaria cardamomum* L., belonging to the *Zingiberaceae* family, referred to as the “Queen of Spices” and holds the distinction of being the second most important “national spice” of India [8,9]. Traditionally, CAR has been employed in the treatment of various ailments, including gum infections, asthma, cataracts, and cardiac, digestive, and renal disorders [8,10,11]. Furthermore, it is noted for its anti-diabetic, anti-inflammatory, antioxidant, and anticarcinogenic properties [12,13]. However, only one study has investigated the effects of an ethanol extract of CAR on cognitive performance in mice. The results indicated that the extract significantly improved memory and learning abilities based on an elevated plus maze test and passive avoidance apparatus [14].

Previous biological studies have predominantly concentrated on the volatile oil derived from CAR, which is known to contain significant components such as terpenes, esters, and flavonoids [13]. The primary constituents identified in CAR volatile oil include both 1,8-cineole and α-terpinyl acetate [15]. Additionally, it has been noted that the organic fractions of CAR extracts encompass various flavonoid compounds, including luteolin, quercetin, kaempferol, and pelargonidin [16]. Nevertheless, there is a lack of data regarding the association between CAR extracts in combination with orlistat. This study aimed to evaluate the effect of orlistat in combination with CAR extract on learning, memory, anxiety, and neuroinflammation in obese mice.

## 2. Material and Methods

### 2.1. Animals and Experimental Design

A total of thirty-one male C57BL/6 mice, each six weeks old and the average weight was of 24 g, were subjected to a one-week acclimatization phase. These mice were then housed in polycarbonate cages maintained at a temperature of 21–23 °C and humidity levels ranging from 35% to 70%, under a controlled 12 h light/dark cycle. They had unrestricted access to tap water and a standard diet. The experimental design included two dietary groups: the control group, which received a control diet (CD) composed of 65% carbohydrates, 25% proteins, and 10% fats, and the cafeteria diet (CAF) group, which consisted of 38.1% fat, 16.6% protein, and 35.5% carbohydrates [17]. This dietary intervention lasted for 10 weeks to induce obesity, after which the mice were divided into four treatment groups for an additional four weeks, with each group containing four mice, as illustrated in Figure 1.

Blood samples were obtained from the retro-orbital vein to assess total cholesterol (TC), low-density lipoprotein (LDL), high-density lipoprotein (HDL), and triglyceride (TG), while the hippocampus was extracted for TNF-α measurement. All animal procedures received approval from the Ethics Committee of Applied Science Private University, adhering to the National Institute of Health Guide for the Care and Use of Laboratory Animals (2023-PHA-30, 6 August 2023).

### 2.2. Obesogenic Mice Diet

The standard diet contained an energy value of 3.57 kcal/g, composed of 65% carbohydrates, 25% proteins, and 10% fats. As previously described and to simulate the human obesogenic diet, animals were provided commercially with human snacks, the cafeteria diet was intentionally designed to reflect the variety and energy density typical of a human obesogenic diet, featuring commercially available snacks with an average caloric density of 4.84 kcal/g, which includes 35.3% carbohydrates, 38.1% fats, and 16.6% proteins [17]. Furthermore, a 10% sucrose solution was provided in addition to clean water. Daily food intake was meticulously recorded, and any spillage was accounted for to ensure a precise assessment of consumption. Body weight (BW) was measured weekly using a digital balance with a precision of ±0.01 g (Kern 440, KERN and SOHN, Balingen, Germany), and length was measured to calculate the Lee obesity index (LOI) for the induction of obesity [18].$$\mathrm{Lee}\;\mathrm{obesity}\;\mathrm{index}\;\%=\frac{\sqrt[{2}]{body\;weight\left(g\right)}}{nas-anal\;lenght}\times 100$$

### 2.3. CAR Water Extract and Drug Preparation

Cardamom (*Elettaria cardamomum*) was obtained from a local market from Amman, Jordan, and its formal identification was confirmed by Dr. Ahmad Khaled, a specialist in botany. The identification specimen number is 1438”. The *Elettaria cardamomum* water extract was prepared by powdering the locally purchased CAR, and then 250 g of powder was ground, soaked in 2.5 L distilled water for 48 h, filtered, and dried to a residue, which was then administered at 500 mg/kg daily [19]. The orlistat drug was dissolved in 0.2% tween-80 solution and administered using oral gavage at 10 mg/kg/day for four weeks [20]. The dose was calculated as follows:Formula: Dose (mg) = 10 mg/kg × BW (kg)

### 2.4. Intraperitoneal Glucose Tolerance Test (IPGTT)

After a period of 10 weeks of induced obesity, an intraperitoneal glucose tolerance test (IPGTT) was conducted on all mice following a 12 h overnight fasting period. The mice were administered an intraperitoneal injection of 0.5 g/kg of BW to assess their glucose tolerance. Blood samples were collected from the tail vein, and glucose concentrations were assessed immediately at 30, 60, and 120 min after the injection [21].

### 2.5. Determination of Serum Lipid Profile and Inflammation in the Hippocampus

Serum levels of TC, TG, LDL, and HDL were evaluated in plasma using the enzyme-linked immunosorbent assay (ELISA) method. This was conducted utilizing commercial mouse kits from BioSystems S.A. in Barcelona, Spain, as well as the Thermo Fisher Scientific ELISA Kit for TNF-α from Waltham, MA, USA, following the provided instructions. Each sample was analyzed in duplicate.

### 2.6. Behavioral Analysis

#### 2.6.1. Novel Object Recognition (NOR) Test

The evaluation of declarative memory was performed through the NOR test, which capitalizes on the tendency of animals to explore a new object more than one they have previously encountered in a familiar setting. In this methodology, the mice subjects were placed in the same apparatus used for the open-field test, located in a dimly lit environment, as described in [22]. The animals underwent a habituation phase within the apparatus, devoid of any stimuli, lasting 20 min each day over four days. This phase aimed to mitigate any anxiety-related behaviors that might emerge from being in an unfamiliar environment, which could potentially affect their performance in the NOR test. The data analysis utilized several metrics: the global habituation index to assess overall exploration levels; the preference index to indicate cognitive function; the discrimination index (DI) to measure memory performance; and the recognition index to evaluate the ability to identify novel objects.

#### 2.6.2. Open-Field Test

The open-field test was utilized to evaluate the locomotor activity of the subjects. Assessing locomotor activity is crucial, as fluctuations in this activity can affect the outcomes of the object recognition test. In brief, the mice subjects were placed in the upper left quadrant of a 50 × 50 cm open field, which was surrounded by walls that were 20 cm high. The time spent moving within the apparatus was recorded. The subjects were permitted to explore the apparatus for 5 min, and the percentage of time spent in motion was calculated regarding the total duration of the test [23].

#### 2.6.3. Y Maze Test

The assessment of working memory and exploratory behavior was conducted utilizing a Y maze apparatus. Each mouse was initially positioned in the central zone. The number of entries into the arms and the frequency of alternations were documented over 10 min using the EthoVisionXT video imaging system. Working memory was evaluated by calculating the ratio of correct alternations to the total number of new arm entries, as outlined in a prior study [24].

#### 2.6.4. Marble-Burying Test

The MBT is believed to indicate repetitive and perseverative behaviors, which may be associated with compulsive tendencies and/or anxiety disorders. This assessment was conducted following the methodology outlined by Deacon and colleagues, with some modifications. In summary, individual mice were placed in a transparent plastic enclosure that contained approximately 5 cm of lightly compressed wood chip bedding to create a flat surface. Fifteen glass marbles, each with a diameter of 1.5 cm, were systematically positioned on the surface, spaced approximately 4 cm apart, forming five rows of three marbles each. During the 10 min evaluation, the latency to initiate digging (defined as the mouse using both front and hind paws to dig the bedding for more than one second), the total number of digging episodes, and the count of marbles buried to two-thirds of their depth were recorded manually [25].

### 2.7. Statistical Analysis

Statistical analysis was performed using the Statistical Package for the Social Sciences (SPSS), version 27.0 for Windows (Chicago, IL, USA). Dependent samples t-tests were employed to evaluate the mean differences within each group over time. Additionally, independent samples t-tests were applied to compare the means of different groups at specific intervals. To determine the statistical significance of the observed differences, the *p*-values associated with these tests were carefully analyzed. Notably, the threshold for determining statistical significance was established at 0.05.

## 3. Result

In Figure 2A, the mean BW of the CAF group significantly increased after 10 weeks compared to the CD group (*p* < 0.001). However, after 4 weeks of treatment, the CAF-ORL group had significantly reduced BW compared to the CAF group, and there was a significant decrease in BW in the CAF-ORL-CARD group compared to the CAF-ORL group (*p* < 0.01). As shown in Figure 2B, the LOI was significantly increased in the CAF group compared to the CD group. Additionally, the LOI was observed in the CAF-ORL-CARD group when compared to the CAF and CAF-ORL groups (*p* < 0.01).

The initial IPGTT assessment revealed significant differences among the tested groups (*p* < 0.01). The peak blood glucose level was recorded 30 min post-glucose administration in the CAF group compared to the CD group (*p* < 0.01), subsequently returning to near baseline levels by 120 min. Notably, the inclusion of CAR in the diet of obese mice led to a significant reduction in blood glucose levels at the 60 min mark, which was, thereafter, in contrast to the CAF-fed mice. The CAF-ORL-CARD group also showed significant reductions in blood glucose levels compared to the CAF-ORL group, as shown in Figure 3.

Table 1 presents the plasma levels of TC and TG across all groups of mice. The concentrations of TC and TG were markedly elevated in mice fed CAF when compared to those in a CD group (*p* < 0.01) and elevated in CAF-ORL compared to CAF-ORL-ORL (*p* < 0.01); however, no significance was found in the CAF-CARD group when compared to the CAF-ORL group (*p* < 0.01). The concentration of TNF-α was significantly higher in the CAF group compared to the CD mice (*p* < 0.01). Conversely, CAR supplementation in the CAF reduced TNF-α levels (*p* < 0.05) when compared to the CAF-CARD group (120.37 ± 10.54 vs. 98 ± 12.17; *p* < 0.01, respectively). Also, the level of TNF-α in the CAF-ORL-CARD group was lower than in the CAF-CARD and CAF-ORL groups (*p* < 0.01).

Figure 4 illustrates the neurobehavioral test parameters across the various experimental groups. In the NOR test, the CAF group exhibited a notable decrease in the discrimination index (DI) compared to the CD group (0.13 ± 0.05 vs. 0.20 ± 0.087; *p* < 0.01), indicating a deficit in recognition memory. Notably, following CAR supplementation, the CAF-CARD group demonstrated a significant increase in DI relative to the CAF group (0.19 ± 0.05 vs. 0.13 ± 0.025; *p* < 0.01). Additionally, the DI for the CAF-ORL-CARD group was higher than that of the CAF-CARD and CAF-ORL groups, as depicted in Figure 4A.

Regarding locomotor activity assessed through the open-field test, Figure 4B reveals a significant difference in the total distance traveled over 10 min between the CAF and CAF-CARD groups (*p* < 0.01) and between the CAF and CAF-ORL-CARD groups (*p* < 0.01). In the Y maze test, a significant difference was found in the novel percentage between the CD and CAF groups (*p* < 0.01), while no significant change was observed following CAR supplementation (*p* > 0.05), or for orlistat and the combination (*p* > 0.05), as shown in Figure 4C. Lastly, the MBT, which evaluates anxiety levels in mice through their digging behavior, indicated that CAF mice buried 60% of the marbles compared to 40% in the CD group (*p* < 0.01). However, this percentage decreased when comparing the CAF group with the CAF-CARD group (*p* < 0.001), and in the CAF-ORL-CARD group compared to the CAF-CARD group (*p* < 0.01), as illustrated in Figure 4D.

## 4. Discussion

In this study, we documented increases in the intake of calorie-dense, highly palatable, and unhealthy foods, such as the CAF, characterized by its high sugar and fat content [26,27,28] This high intake has increased the prevalence of obesity, which can potentially cause hippocampal and white adipose tissue inflammation, lead to cognitive impairment, and elevate the risk of developing psychiatric disorders, particularly depression and anxiety disorders [29,30]. Zeeni’s study on 30 mice found that the CAF caused weight gain; increased energy intake, serum TC levels, insulin levels, and glucose intolerance; and caused heart, kidney, and liver damage compared to other diets [31]. Additionally, the CAF caused significant changes in weight, abdominal fat, and higher IL-6 levels [32]. Lewis et al. showed that rats fed CAF for five weeks consumed more food, had higher energy intake, larger fatty livers, glucose intolerance, hyperinsulinemia, elevated interleukins 6, 10, 12, TNF-α, TC, TG, and reduced HDL [33]. The obesity induced by the CAF led to cognitive impairments in the hippocampus. The MWM test revealed decreased long-term retention memory, and the NOR tasks demonstrated deficits in recognition memory in CAF rats [34].

Consistent findings were observed in our present study. The average weight of the CAF group demonstrated a statistically significant increase after ten weeks of feeding (*p*-value < 0.001). Moreover, the LOI and TC, LDL, and TG levels were notably elevated in mice subjected to a CAF (*p*-value < 0.01). Likewise, the concentration of TNF-α was significantly higher in the CAF group (*p*-value < 0.01) [35,36].

Orlistat acts as a powerful and irreversible lipase inhibitor, which blocks TG absorption from the diet [36]. In a study of 72 obese patients diagnosed with gout, the group who received 120 mg orlistat showed a significant decrease in LDL levels with an average weight reduction of 2.85 kg. In contrast, the placebo group reported a lower average weight loss of only 0.76 kg [37]. The administration of 120 mg of orlistat three times daily with a hypocaloric diet for 16 weeks promoted weight loss in obese alcoholic fatty liver patients compared to the control, with significantly improved insulin sensitivity and liver enzymes. Also, orlistat enhanced TNF-α levels compared to the control. However, decreasing serum TNF-α levels did not significantly affect serum IL-6 levels [38].

A study conducted by Othman et al. showed that after six weeks of inducing obesity by HFD with 31% of energy from fat, an orlistat dose of 10 mg/kg/day led to significant reductions in weight (202.40 ± 11.80 gm vs. 161.70 ± 10.23 gm, *p*-value < 0.05, respectively), abdominal circumference (18.04 ± 0.35 mm vs. 16.69 ± 0.41 mm, *p*-value < 0.05, respectively), improved TC and TG. Additionally, it increased the anti-inflammatory marker while decreasing the pro-inflammatory marker (IL-10 immunoexpression) [39]. In the current study, after four weeks of treatment, orlistat significantly reduced mean BWs compared to the CAF group. Orlistat significantly decreased BW (*p*-value < 0.01) and the LOI when combined with CAR (*p*-value < 0.01). Indeed, orlistat reduced TC, TG, LDL, fasting blood glucose, and the cardiac risk ratio (0.43 ± 0.10 vs. 1.29 ± 0.2, *p*-value < 0.05, respectively), and increased HDL compared to the CAF group.

CAR is a promising natural antioxidant and anti-inflammatory plant due to its high content and variety of polyphenols [40,41]. Supplementing the diet with 3 gm of CAR powder has shown promising results in reducing blood glucose levels in obese rats compared to those on a rich sugar-fat diet [10]. It also effectively reduces TNF-α and IL-6 [39,40]. In our study, CAR supplementation significantly improved glucose tolerance. The CAF-CARD group showed no considerable plasma levels of LDL and TG compared to the CAF-ORL group. However, CAR did reduce lipid parameters compared to the CAF. Adding CAR to the regimen of the CAF significantly reduced TNF-α levels compared to the CAF-ORL group. These findings highlight the promising potential uses of CAR in effectively managing obesity-related parameters and decreasing inflammation.

CAR oil therapy has been shown to improve the degenerative alterations observed in the hippocampus and cortical areas of the brain, attributed to its neuroprotective properties. Research indicates that the accumulation and deposition of amyloid beta (Aβ) play a significant role in the development of Alzheimer’s disease, resulting in neurodegeneration [42].

The cafeteria diet has been linked to cognitive decline, learning problems, and working memory deficits due to adipocyte hypertrophy and hyperplasia that lead to chronic inflammation [43]. The low-grade systemic inflammation induced by obesity, associated with high fat intake, can lead to neuroinflammation by compromising the integrity of the blood–brain barrier, promoting insulin resistance [44,45], and causing mitochondrial impairment in the cerebral cortex [30,46]. The neuroinflammatory alterations related to obesity occur in the cerebellum, hippocampus, hypothalamus, and cerebral cortex, resulting in memory deficits and elevated expression of inflammatory genes [47]. In the hippocampus and cerebral cortex, prolonged exposure to fat-rich diets, such as the CAF, increased TNF-α, IL-6, IL-1, and leptin levels. Additionally, microglial activation was observed in the prefrontal cortex and hippocampus [47,48]. Furthermore, the cafeteria diet increased Iba-1 expression and impaired recognition memory in obese Wistar rat brains [48].

Medicinal plants from the Zingiberaceae family (e.g., curcumin and ginger) that contain polyphenols and share antioxidant activity and memory enhancement properties with CAR have been shown to improve the effectiveness of treatments for obesity and related metabolic disorders [49]. Extracts from curcumin significantly lowered the gene expression of TNF-α and IL-6 [50]. The administration of 6-shogaol, a bioactive compound in ginger, significantly reduced microgliosis and astrogliosis in mice injected with intrahippocampal AβO [51], suppressed the phosphatidylinositol 3-kinase (PI3K)/AKT/mTOR signaling pathway [52], and ginger extract at concentrations up to 400 µg/mL resulted in decreased reactive oxygen species production [15].

A study by Khedr et al. investigated the effect of 1200 mg of cinnamon as a natural lipase inhibitor in 80 participants with obesity for 60 days. The orlistat and cinnamon groups experienced a significant reduction in BMI and TGs. In another study, the administration of cinnamon supplementation showed a significant reduction in serum lipase levels compared to the control and orlistat-supplemented group [53].

However, following extensive search across multiple databases, we identified a significant research gap in comparative studies examining the efficacy of administering orlistat and CAR with the CAF in addressing effects on locomotor activity, working memory, exploration, and repetitive and perseverative behaviors. This highlights the importance of our research. Our results reveal a significant decrease in DI, anxiety level, and locomotor activity in the CAF group compared to the CD group. Conversely, the combination of orlistat and CAR demonstrated a significantly higher increase in DI, anxiety level, and locomotor activity compared to single-level management. However, no significant change was observed following CAR supplementation, orlistat, or the combination in working memory and exploratory behavior.

The findings suggest that orlistat, as a lipase enzyme inhibitor, can effectively mitigate the effects of the CAF. At the same time, CARs may have a positive influence due to their antioxidative properties. Still, additional research is required to determine the long-term impacts and possible adverse reactions.

Several limitations were identified in our study: Firstly, all mice were male, making these results difficult to generalize to the female gender. Females exhibit distinct metabolic and physiological reactions to CAF when compared to males, as noted in previous research [29]. Interestingly, this has led to inconsistencies in the observed gender differences regarding the effects of caffeine on hippocampal memory. Furthermore, alongside the variations in caffeine’s impact on aversive behavior tests, both male and female mice consistently demonstrate gender-specific differences in their physiological and behavioral responses to acute stressors [30]. Secondly, a significant concern is that the limited sample size may have compromised the statistical power necessary to identify small effect sizes. To address this issue, future studies should consider larger sample sizes to enhance the detection of small effects and improve the overall reliability and validity of the findings, and future research should include both male and female subjects to ensure that the results are more comprehensive and reflective of gender-specific variations in response to CAF and neuroinflammation.

## 5. Conclusions

The current study showed that the beneficial role of orlistat in combination with CAR has synergistic effects on neuroinflammation, spatial memory, and recognition memory, suggesting that orlistat may offer potential benefits for cognitive impairment and hippocampal dysfunction associated with white adipose tissue inflammation inherent to obesity. These effects could be explained by the synergistic anti-inflammatory activities of orlistat and CAR combination on cafeteria diet obesity-induced mice.

## Figures and Tables

**Figure 1 medicina-61-00263-f001:**
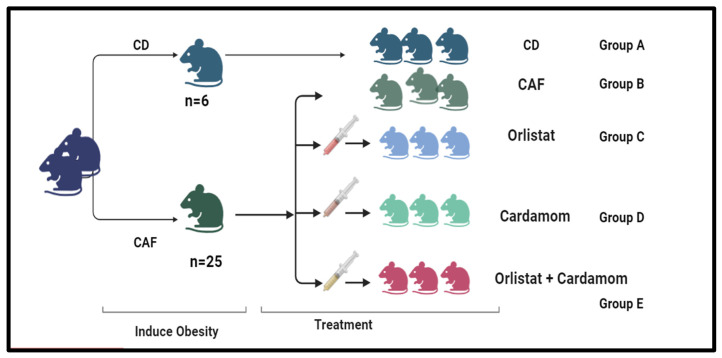
Group 1 was fed with a cafeteria diet (CAF) for 10 weeks to induce obesity. Group 2 served as the control diet (CD) group, representing a standard dietary regimen. Each group was further subdivided into subgroups: Group A, which acted as the CD, was provided with regular mice chow throughout the study without any treatment. Group B was designated as the CAF group. Group C received CAF with orlistat at a dosage of 10 mg/kg of body weight (BW) for a period of 4 weeks (CAF-OR). Group D was administered CAF along with CAR at a dosage of 500 mg/kg for 4 weeks. Lastly, Group E was administered CAF in conjunction with both orlistat (10 mg/kg of BW) and CAR (500 mg/kg) for 4 weeks (CAF-OR-CARD).

**Figure 2 medicina-61-00263-f002:**
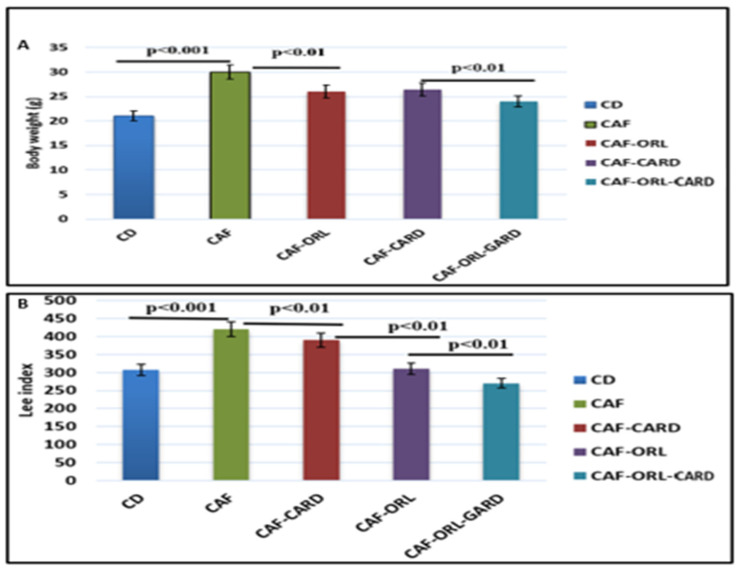
The change in BW and Lee obesity index during the study. The BW (**A**) and LOI (**B**). (**A**) BW (g/) for 10 weeks. Mice were fed standard chow as a control diet, a cafeteria diet to induce obesity (CAF), CAF and orlistat (10 mg/Kg BW), CAF and CAR (500 mg/kg) extract, or CAF with orlistat and CAR. (**B**) Rhe LOI of mice fed cafeteria diet after 4 weeks of obesity induction in the cafeteria group compared to the control group (CD) fed standard rat chow. Data represent the mean ± standard error of the mean (mean ± SEM) (n = 6 rats per group). Values are significantly different when *p* < 0.05. CD, control diet; CAF, cafeteria diet; CAF-CARD, cafeteria diet with CAR; CAF-ORL, cafeteria diet with orlistat; CAF-ORL-CARD, cafeteria diet with orlistat and CAR.

**Figure 3 medicina-61-00263-f003:**
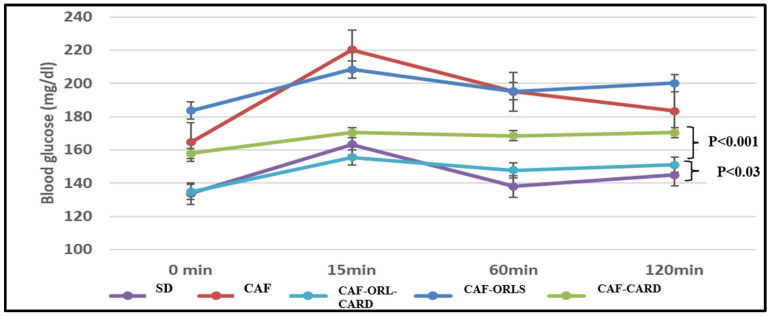
Blood glucose concentrations during intraperitoneal glucose tolerance test in C57BL/6 male mice. Each value represents the mean ± standard error of the mean (mean ± SEM) for 6 mice/group. Values are significantly different when *p* < 0.05. SD, control diet; CAF, cafeteria diet; CAF-CARD, cafeteria diet with CAR; CAF-ORL, cafeteria diet with orlistat; CAF-ORL-CARD; cafeteria diet with orlistat and CAR.

**Figure 4 medicina-61-00263-f004:**
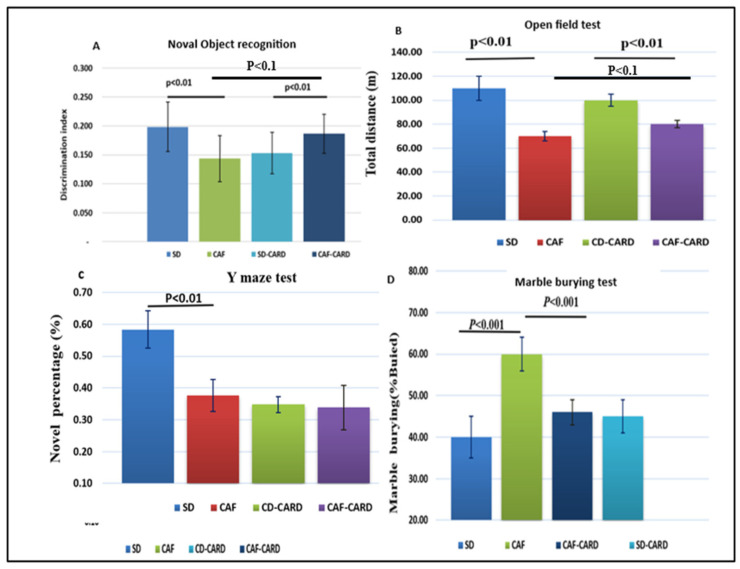
The impact of CAR on locomotor activity and the performance of mice in various behavioral tests, including the object recognition test, Y maze test, and marble-burying test, was examined. (**A**) The discrimination index. (**B**) The locomotor activity. (**C**) Y maze test. (**D**) Marble-burying behavior. The results are presented as the mean ± SEM for six animals per group.

**Table 1 medicina-61-00263-t001:** Plasma lipid profile and neuroinflammation in mice.

Groups	CD	CAF	CAF-CARD	CAF-ORL	CAF-ORL-CARD
LDL (mg/dL)	51.40 ± 12.94 *^a^	121.027 ± 24.44 *^b^	110.027 ± 24.0	111.027 ± 25.44 *^c^	89
TC (mg/dL)	61.03 ± 11.61 *^a^	114.48 ± 21.42 *^b^	104.48 ± 22.42	100.40 ± 21.42 *^c^	83 ± 11.43
HDL (mg/dL)	45 ± 6.06 *^a^	72.43 ± 12.52 *^b^	62.42 ± 12.52	61.43 ± 12.5	60 ± 32.0
TG (mg/dL)	62.81 ± 17.80	87 ± 20.01 *^b^	62.11 ± 17.40	60.81 ± 14	61.81 ± 17.80
Fasting blood glucose (mg/dL)	121 ± 10.4 *^a^	155 ± 12.43 *^b^	130 ± 16.0	133 ± 15.0 *^c^	120 ± 11.2
Atherogenic index ^ψ^	0.28 ± 0.10 *^a^	0.53 ± 0.16 *^b^	0.43 ± 0.10	0.42 ± 0.11 *^c^	0.33 ± 0.023
Cardiac risk ratio ^¥^	0.56 ± 0.17 *^a^	1.29 ± 0.21 *^b^	0.44 ± 0.10	0.43 ± 0.10 *^c^	0.29 ± 0.01
TNF-α (ng/L) in hippocampus	80.89 ± 20.14 *^a^	120.37 ± 10.54 *^b^	99 ± 12.17	98 ± 12.17	82 ± 21

Data are presented as mean ± standard error of the mean (mean ± SEM) and are significant at *p* < 0.05 as follows: (*^a^) when comparing CD with CAF, (*^b^) when comparing CAF with CAF-CAFD, (*^c^) when comparing CAF-CARD with CAF-CARD. TC, total cholesterol; TG, triglyceride; HDL, high-density lipoprotein; LDL, low-density lipoprotein; CD, control diet; CAF, cafeteria diet; CAF-CARD, cafeteria diet with CAR; CAF-ORL, cafeteria diet with orlistat; CAF-ORL-CARD, cafeteria diet with orlistat and CAR. ^Ψ^ Atherogenic index: cholesterol-HDL-C)/HDL-C. ^¥^ Cardiac risk ratio: = [TC/HDL-c].

## Data Availability

All data used in the study are available upon request from the corresponding author.

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
