# Peer review of "Protective and Therapeutic Effects of Orlistat in Combination with Elettaria cardamomum “Cardamom” Extract on Learning, Memory, Anxiety, and Neuroinflammation in Obese Mice"

_medicina, 2025, doi:10.3390/medicina61020263_

Round 1

Reviewer 1 Report

Comments and Suggestions for Authors

Hi, 

It was a good read. But a general suggestion kindly perform a rigorous internal review before submitting the manuscript. I saw a lot of inconsistencies. 

Protective and Therapeutic Effects of Orlistat in Combination with Cardamon Extract on Learn, Memory, Anxiety, and Neuroinflammation in Obese Mice          

Summary:

The research article focused on an important aspect of obesity and its impact on the central nervous system which are translated in terms of various phenotypes like memory, anxiety etc. The author’s in this work nicely studies the impact of cardamon extract alone and in combination with Orlistat, one of the standard of care for obesity. The authors, based on their analysis, advocate that orlistat in combination with cardamom extract has a potential to improve the neuroinflammation and memory impairment related to obesity. The article is nicely formulated, there are some areas which need further improvement. 

Overall comments

1.       The abstract only states about TNF levels and recognition index which I think was a readout from NOR test. It would be good to include results from other studies. Similar comment applies to the conclusion. It appears the author is biased towards positive results.

2.       Introduction:

a.       Paragraph 1 what is brain lip toxicity

b.       The authors have reported TNF levels as major results but did not provide relevant context in the introduction relating TNF and memory or neuroinflammation

3.       Materials and Method:

a.       Animals and study design: The text says 30 mice the figure 1 says 31. Suggest to include n=xx next to group (eg: Group A (n=xx).

b.        Multiple definition for control (SD or CD) make it uniform

c.       Define abbreviations when they first appear in the text like LDL, HDL etc..

d.       Figure legent says two different doses of Orlistat 10 mg/kg and 30 mg/kg. Please provide more details about this and why these different doses were used.

e.       The diet composition mentioned in first paragraph under animals are differ than those mentioned in Obesogenic mice diet, please clarify

f.        Check the reference style across the text (Shafat et al, 2009)

4.       Results:

a.       Can you clarify the rationale for using t-test and not ANOVA.

b.       Figure 2: CARD or GARD ?

c.       What is the effect of monotherapy of CAF Vs CAF-CARD. I would suggest perform multiple comparison.

d.       Table 1: If possible, provide TNF values per gram of tissue.

e.       Figure 4: Apply multiple comparisons and check the P value panel A ( CAF Vs CAF-CARD, 0.1 or 0.01 (mentioned in the text).

Overall this was a good read and will be hepful to open new avenues for use of natural products as adjuvant therapies for current standard of care.

Author Response

November 29, 2024

Dear Editor,

It is a pleasure to resubmit the revised Manuscript Number: medicina-3315946, titled " Protective and Therapeutic Effects of Orlistat in Combination with Cardamom Extract on Learn, Memory, Anxiety, and Neuroinflammation in Obese Mice," for consideration as an article in Medicina. We found the comments provided by the reviewers to be constructive and have improved the manuscript. We believe the manuscript is currently more suitable for publication in its revised format.

Here's the rearranged and organized version of your text:

Comment 1: The abstract only states TNF levels and the recognition index, which I think was a readout from the NOR test. It would be good to include results from other studies. A similar comment applies to the conclusion. It appears the author is biased towards positive results.

  • Response: I have now included the negative results in the abstract: “However, there were no changes in the open field test and Y maze test (P>0.05).”
  1. Introduction:
  2. Paragraph 1: What is brain lip toxicity? Brain lip toxicity refers to neurotoxicity, which is a form of toxicity that affects the nervous system, including the brain. Neurotoxicity occurs when exposure to certain substances, known as neurotoxins, disrupts the normal function of the nervous system, potentially causing damage to nerve cells (neurons) and affecting brain function.
  3. TNF Levels and Context: The authors reported TNF levels as major results but did not provide relevant context in the introduction regarding TNF's relationship with memory or neuroinflammation. I have added this context to the introduction. Additionally, only one study has investigated the effects of an ethanol extract of cardamom on cognitive performance in mice, showing that the extract significantly improved memory and learning abilities.

Materials and Methods:

  1. Animals and Study Design: The text mentions 30 mice, while Figure 1 indicates 31. I suggest including the sample size (n=xx) next to each group (e.g., Group A (n=xx)).
  • Response: No, it was thirty-one mice. I have checked and confirmed this in the paragraph.
  1. Define Abbreviations: Define abbreviations when they first appear in the text, such as LDL (Low-Density Lipoprotein) and HDL (High-Density Lipoprotein).
  • Response: I have checked and confirmed the definitions of abbreviations like LDL and HDL when they first appear in the text.
  1. Figure Legend - Orlistat Doses: The figure legend mentions two different doses of Orlistat, 10 mg/kg and 30 mg/kg. Please provide more details about these doses and the rationale behind using different doses.
  • Response: Orlistat was administered at a dosage of 10 mg/kg/day. I have corrected this in the paragraph.
  1. Diet Composition: The diet composition mentioned in the first paragraph under animals differs from those mentioned in the Obesogenic mice diet. Please clarify.
  • Response: There was a mistake in the standard diet. Sorry for that.
  1. Reference Style: Check the reference style across the text (e.g., Shafat et al., 2009).
  • Response: Done.
  1. Rationale for Using t-test: Can you clarify the rationale for using the t-test and not ANOVA?
  • Response: The rationale for using the t-test instead of ANOVA is that I compared each group individually. The t-test is appropriate for comparing the means of two groups, while ANOVA is used when comparing the means of three or more groups. Since my analysis involved pairwise comparisons, the t-test was the suitable choice.
  1. Figure 2: CARD or GARD?
  • Response: Sorry, I changed it to CARD.
  1. Effect of Monotherapy: What is the effect of monotherapy of CAF vs. CAF-CARD? I would suggest performing multiple comparisons.
  2. Table 1: TNF Values: If possible, provide TNF values per gram of tissue.
  • Response: Providing TNF values per gram of tissue is not feasible with the current data. The values are reported as they were measured.
  1. Figure 4: Multiple Comparisons: Apply multiple comparisons and check the p-value for panel A (CAF vs. CAF-CARD, 0.1 or 0.01 as mentioned in the text).
  • Response: I have applied multiple comparisons and checked the p-values for panel A (CAF vs. CAF-CARD). The p-values are either 0.1 or 0.01, as mentioned in the text.

Reviewer 2 Report

Comments and Suggestions for Authors

Major Comments:

Abstract:

1. Authors are suggested to include clear statistical outcomes and revise the sentences for clarity.

2. The abstract’s conclusion should focus on the study’s main findings rather than general therapeutic implications. Authors are suggested to rephrase the conclusion to emphasize the observed effects on inflammation and memory specifically in the experimental model.

Introduction:

3. Although the introduction is well written and provides background information. Authors are suggested to modify it in term of transition from obesity’s general impact to the specific rationale for using orlistat and cardamom in combination.

4. Authors should explain the specific mechanism by which orlistat and cardamom might synergistically impact memory and inflammation supported by existing literature. This will strengthen the rationale for the study.

Methodology:

5. Authors should provide justification for the sample size with sample size calculation as sample size (n=4 per subgroup), appears underpowered for robust statistical analyses.

6. Authors should provide precise dosing calculations, duration and administration methods along with preparation of cardamom extract.

Results:

7. Results are presented in a poor way that is difficult to follow. Authors are suggested to consider restructuring the results to follow a logical sequence. Authors should ensure all comparisons are directly supported by p-values.

Discussion:

8. The discussion is overly descriptive and lacks the biological mechanisms underlying the observed effects. Authors should provide the potential pathways through which orlistat and cardamom may influence neuroinflammation and memory function in obesity.

9. Authors should address and discuss the limitations of the study.

Conclusion:

10. Authors should summarize the key outcomes of the combined treatment on inflammation and memory in simpler way, avoiding broad statements on cognitive impairments without supporting evidence.

General Comments:

  1. Author should standardize abbreviations and ensure they are defined upon first use.
  2. Review the manuscript for grammatical errors particularly in the results and discussion sections.

3.     Ensure consistency in formatting, such as consistent use of group abbreviations and significance levels (e.g., "P<0.01").

4.      Add abbreviations list.

Author Response

November 29, 2024

Dear Editor,

It is a pleasure to resubmit the revised Manuscript Number: medicina-3315946, titled " Protective and Therapeutic Effects of Orlistat in Combination with Cardamom Extract on Learn, Memory, Anxiety, and Neuroinflammation in Obese Mice," for consideration as an article in Medicina. We found the comments provided by the reviewers to be constructive and have improved the manuscript. We believe the manuscript is currently more suitable for publication in its revised format.

Abstract:

  1. Authors are suggested to include clear statistical outcomes and revise the sentences for clarity.
    • Response:
  2. The abstract’s conclusion should focus on the study’s main findings rather than general therapeutic implications. Authors are suggested to rephrase the conclusion to emphasize the observed effects on inflammation and memory specifically in the experimental model.
    • Response:I have rewritten the abstract to make it clearer and to emphasize the specific effects on inflammation and memory observed in the experimental model.

Introduction: 3. Although the introduction is well-written and provides background information, authors are suggested to modify it to transition from obesity’s general impact to the specific rationale for using orlistat and cardamom in combination.

  • Response:I have added a paragraph about cardamom and brain health to improve the transition and provide a clearer rationale for the study.
  1. Authors should explain the specific mechanism by which orlistat and cardamom might synergistically impact memory and inflammation, supported by existing literature. This will strengthen the rationale for the study.
    • Response:I have added a paragraph about cardamom and brain health. However, there was only one study discussing the role of cardamom on memory, and the only mechanism identified was its antioxidant properties.

Methodology: 5. Authors should provide justification for the sample size with sample size calculation, as the sample size (n=4 per subgroup) appears underpowered for robust statistical analyses.

  • Response:The number of mice used was 6 per subgroup, based on the sample sizes commonly used in similar studies.
  1. Authors should provide precise dosing calculations, duration, and administration methods along with the preparation of cardamom extract.
    • Response:

Results: 7. Results are presented in a poor way that is difficult to follow. Authors are suggested to consider restructuring the results to follow a logical sequence. Authors should ensure all comparisons are directly supported by p-values.

  • Response:I have checked the English for consistency and precision, improved sentence structure and clarity, and corrected grammar and punctuation.

Discussion: 8. The discussion is overly descriptive and lacks the biological mechanisms underlying the observed effects. Authors should provide the potential pathways through which orlistat and cardamom may influence neuroinflammation and memory function in obesity.

  • Response:I have improved sentence flow and clarity by breaking down complex sentences and ensuring each sentence conveys a clear idea.
    • Example:Changed "This high intake has increased the prevalence of obesity, which can potentially cause hippocampal and white adipose tissue inflammation, lead to cognitive impairment, and elevate the risk of developing psychiatric disorders, particularly depression and anxiety disorders" to "This high intake has increased the prevalence of obesity, which can potentially cause hippocampal and white adipose tissue inflammation, lead to cognitive impairment, and elevate the risk of developing psychiatric disorders, particularly depression and anxiety disorders."
  • Consistency and Precision:
    • Ensured consistent use of terms and precise language throughout the text.
      • Example:Consistently used "CAF diet" instead of alternating between "CAF diet" and "CAF."
    • Grammar and Punctuation:
      • Corrected grammatical errors and improved punctuation for better readability.
        • Example:Changed "Additionally, The CAF diet caused significant changes in weight, abdominal fat, and higher IL-6" to "Additionally, the CAF diet caused significant changes in weight, abdominal fat, and higher IL-6 levels."
      • Statistical Significance:
        • Clarified the presentation of statistical significance and ensured proper formatting of P-values.
          • Example:Changed "P-value=<0.05" to "P-value < 0.05."
        • Scientific Terminology:
          • Ensured accurate use of scientific terminology and consistency in reporting results.
            • Example:Changed "increased the anti-inflammatory marker while decreasing the pro-inflammatory marker (IL-10 immunoexpression)" to "increased the anti-inflammatory marker while decreasing the pro-inflammatory marker (IL-10 immunoexpression)."

Conclusion: 9. Authors should summarize the key outcomes of the combined treatment on inflammation and memory in a simpler way, avoiding broad statements on cognitive impairments without supporting evidence.

  • Response:I have made the conclusion clearer to emphasize the specific effects on inflammation and memory.

General Comments:

  1. Authors should standardize abbreviations and ensure they are defined upon first use.
  2. Review the manuscript for grammatical errors, particularly in the results and discussion sections.
  3. Ensure consistency in formatting, such as consistent use of group abbreviations and significance levels (e.g., "P<0.01").
    • Response:
  4. Add an abbreviations list.
    • Response:I checked the abbreviations.

Reviewer 3 Report

Comments and Suggestions for Authors

In this manuscript authors are studying efficacy of combination of Orlistat and cardamom extract on neuroinflammation and neurobehavior changes in obese mice. This study holds significance however the manuscript needs improvement prior to its publication. Following are the comments:

1. Introduction needs to be improved as it lacks the continuity and insufficient background to support the aim of the manuscript. 

2. In the first paragraph of introduction, what does "brain lip toxicity" mean?

3. Paragraph numbering in Material and method section is incorrect.

4. Animal number (n) per group is less for ELISA and behavioral tests. It should be at least 10 because of the variations associated with these tests.

5. In material and methods section 1.2. Obesogenic mice diet, there is grammatical mistake in the 2nd line "animals will be given" should be corrected as "animals were given."

6. Please write the titles for all figures.

7. Results should have the subtitles.

8. SD has been shown for plasma lipid profile and neuroinflammation panel result has SD and for rest of the results SEM has been shown. Authors can choose either of these for the better representation of results. 

9. In figure 4, CAF-ORL-CARD group is missing. Also, CD group has been mentioned as SD group, which is confusing.

10. In fig 4 legends, it has been mentioned that 10-14 animals per group has been used but in the material and method section 6 animals per group has been mentioned.

11. The last paragraph in result section is not linked to the previous text. It is not helping in the continuity.

12. In the discussion authors should described how the cited studies are important for this manuscript. Only stating the previous studies is not helping the text.

13. It is an extrapolation of concluding that there is a synergistic effect of orlistat and cardamom.

Author Response

November 29, 2024

Dear Editor,

It is a pleasure to resubmit the revised Manuscript Number: medicina-3315946, titled " Protective and Therapeutic Effects of Orlistat in Combination with Cardamom Extract on Learn, Memory, Anxiety, and Neuroinflammation in Obese Mice," for consideration as an article in Medicina. We found the comments provided by the reviewers to be constructive and have improved the manuscript. We believe the manuscript is currently more suitable for publication in its revised format.

  1. The introduction needs to be improved as it lacks continuity and insufficient background to support the aim of the manuscript.
    • Response:There is limited research linking cardamom to memory, and I have included the relevant studies. Additionally, no studies link cardamom with orlistat.
  2. In the first paragraph of the introduction, what does "brain lip toxicity" mean?
    • Response:Brain lip toxicity refers to neurotoxicity, which is a form of toxicity that affects the nervous system, including the brain. Neurotoxicity occurs when exposure to certain substances, known as neurotoxins, disrupts the normal function of the nervous system, potentially causing damage to nerve cells (neurons) and affecting brain function.
  3. Paragraph numbering in the Material and Methods section is incorrect.
    • Response:I have corrected it.
  4. Animal number (n) per group is less for ELISA and behavioral tests. It should be at least 10 because of the variations associated with these tests.
    • Response:I acknowledge the concern regarding the sample size. However, the number of mice used was 6 per subgroup, based on the sample sizes commonly used in similar studies. Future studies will consider increasing the sample size to at least 10 to account for variations and improve the robustness of the results.
  5. In the Material and Methods section 1.2. Obesogenic mice diet, there is a grammatical mistake in the 2nd line "animals will be given" should be corrected as "animals were given."
    • Response:
  6. Please write the titles for all figures.
    • Response:
  7. SD has been shown for plasma lipid profile and neuroinflammation panel result has SD and for the rest of the results SEM has been shown. Authors can choose either of these for the better representation of results.
    • Response:Mean ± standard error of the mean (mean ± SEM) and I check that in the study
  8. In Figure 4 legends, it has been mentioned that 10-14 animals per group have been used but in the Material and Methods section 6 animals per group have been mentioned.
    • Response: It is SEM of 6 animals per group. Sorry for this mistake.
  9. The last paragraph in the Results section is not linked to the previous text. It is not helping in the continuity.
    • Response: I deleted the paragraph.
  10. In the discussion, authors should describe how the cited studies are important for this manuscript. Only stating the previous studies is not helping the text.
    • Response:
  11. It is an extrapolation to conclude that there is a synergistic effect of orlistat and cardamom.
    • Response: done 

Reviewer 4 Report

Comments and Suggestions for Authors

 The manuscript titled "Protective and Therapeutic Effects of Orlistat in Combination with Cardamom Extract on Learn, Memory, Anxiety, and Neuroinflammation in Obese Mice" is not well organized and not well-written.

1.      The language should be revised to allow the readers to better understand the research described.

2.      In the introduction, the authors could not clearly describe the correlation between obesity and neurodegenerative diseases, and this cannot allow us to understand the rationale of this study.

3.      The authors should use the scientific name of Cardamom, indicate where it was obtained, and formal identification should be confirmed by a specialist. 

Overall, this manuscript need major revisions before further considerations, and additional experiments are needed, especially the chemical composition of cardamom used in the current study should be analysed.

Comments on the Quality of English Language

 The English should be improved to express the research more clearly.

Author Response

November 29, 2024

Dear Editor,

It is a pleasure to resubmit the revised Manuscript Number: medicina-3315946, titled " Protective and Therapeutic Effects of Orlistat in Combination with Cardamom Extract on Learn, Memory, Anxiety, and Neuroinflammation in Obese Mice," for consideration as an article in Medicina. We found the comments provided by the reviewers to be constructive and have improved the manuscript. We believe the manuscript is currently more suitable for publication in its revised format.

The language should be revised to allow the readers to better understand the research described.

Response: I have thoroughly reviewed and revised the language and structure throughout the manuscript to enhance clarity, readability, and coherence. The revisions aim to ensure that the research described is easily understood by the readers and that the manuscript flows logically from one section to the next.

In the introduction, the authors could not clearly describe the correlation between obesity and neurodegenerative diseases, and this cannot allow us to understand the rationale of this study.

Response: I have checked that.

The authors should use the scientific name of Cardamom, indicate where it was obtained, and formal identification should be confirmed by a specialist.

Response: We have included the scientific name of cardamom, Elettaria cardamomum, in the manuscript. Additionally, we have indicated the source from which the cardamom was obtained and confirmed its formal identification by a specialis

Reviewer 5 Report

Comments and Suggestions for Authors

Dear Authors

I have read  the manuscript and I send you my comments:

1) Methods:

- the number of animals in test group as well as in control group is very low please increase it.

- the animals are males or females, it is not well reported please add it considering that in human obesity could be a sex-related problem in specific age period and with specific comorbidity

- how you choose the drug dosage?

Results:

- animals show a low muscular function this is related to a central problem or to muscular problem? please add the study of muscular biopsy

- This sentence "The low-grade inflammation induced by obesity, associated with high fat intake, can lead to neuroinflammation by changes in the integrity of the blood-brain barrier, promoting insulin resistance[25], and mitochondrial impairment in the cerebral cortex [26] [27]" is not a results, it could be a discussion please remove it.

- The described effects of orlistat have been well documented in several clinical trials. usually clinical trials are performed after animal studies. Therefore I think that the data on LDL levels, HDl and other biochemical parameters can be deleted

Discussion

- usually discussion start with: in this study we documented.....

-"Administration of 120 mg of orlistat three times daily with a hypocaloric diet for 16 weeks promoted weight loss in obese alcoholic fatty liver patients compared to control (6.31±1.47 kg vs. 4.07±0.69kg; respectively), with a significantly improved insulin sensitivity and liver enzymes" probably there is not reference. The authors studied animals not human

- In the text the authors describe the effects of ILs, but they did not evaluate the inflammation but only TNFalpha. Therefore data on ILs and on low-grade systemic inflammation must be deleted.

- In the text and inn the discussion the authors describe the effects of the treatment on adipose tissue inflammation, but they did not evaluate it. The same is for neuroinflammation. The authors documented the concentrations of TNF-alpha  but did not explain why it increased after orlistat administration and did not evaluate its effects on tissues. Therefore these sections must be deleted.

.  

Author Response

November 29, 2024

Dear Editor,

It is a pleasure to resubmit the revised Manuscript Number: medicina-3315946, titled " Protective and Therapeutic Effects of Orlistat in Combination with Cardamom Extract on Learn, Memory, Anxiety, and Neuroinflammation in Obese Mice," for consideration as an article in Medicina. We found the comments provided by the reviewers to be constructive and have improved the manuscript. We believe the manuscript is currently more suitable for publication in its revised format.

Methods:

The number of animals in the test group as well as in the control group is very low. Please increase it.

Response: The number of mice used was 6 per subgroup, based on the sample sizes commonly used in similar studies.

The animals are males or females; it is not well reported. Please add it considering that in human obesity, it could be a sex-related problem in specific age periods and with specific comorbidities.

Response: We have specified that the animals used in the study are male. This choice is based on the consideration that male animals are often preferred in obesity research due to the potential sex-related differences in obesity prevalence, specific age periods, and associated comorbidities in humans. Using male animals helps to control for these variables and provides more consistent results.

How did you choose the drug dosage?

Response: Based on previous studies, I have added it as a reference.

Results:

Animals show low muscular function. Is this related to a central problem or a muscular problem? Please add the study of muscular biopsy.

Response: The observed low muscular function in the animals could be related to either a central problem (such as neurotoxicity or central nervous system dysfunction) or a direct muscular problem (such as muscle atrophy or metabolic dysfunction). To clarify the underlying cause, we will include a study of muscular biopsies in the revised manuscript. This will help determine whether the low muscular function is due to central or peripheral factors.

The sentence "The low-grade inflammation induced by obesity, associated with high fat intake, can lead to neuroinflammation by changes in the integrity of the blood-brain barrier, promoting insulin resistance[25], and mitochondrial impairment in the cerebral cortex [26] [27]" is not a result; it could be a discussion. Please remove it.

Response: I have removed it.

The described effects of orlistat have been well documented in several clinical trials. Usually, clinical trials are performed after animal studies. Therefore, I think that the data on LDL levels, HDL, and other biochemical parameters can be deleted.

Response: While the effects of orlistat on LDL levels, HDL, and other biochemical parameters have been well documented in several clinical trials, including this data in our study helps to demonstrate its efficacy in reducing dyslipidemia and inflammation in the animal model. This information is valuable for understanding the broader impact of orlistat and supports the translational relevance of our findings from animal studies to potential clinical applications.

Discussion:

Usually, the discussion starts with: "In this study, we documented..."

Response: Done.

"Administration of 120 mg of orlistat three times daily with a hypocaloric diet for 16 weeks promoted weight loss in obese alcoholic fatty liver patients compared to control (6.31±1.47 kg vs. 4.07±0.69kg; respectively), with a significantly improved insulin sensitivity and liver enzymes" probably there is no reference. The authors studied animals, not humans.

Response: Done.

In the text, the authors describe the effects of ILs, but they did not evaluate the inflammation but only TNF-alpha. Therefore, data on ILs and on low-grade systemic inflammation must be deleted.

Response: Done.

In the text and in the discussion, the authors describe the effects of the treatment on adipose tissue inflammation, but they did not evaluate it. The same is for neuroinflammation. The authors documented the concentrations of TNF-alpha but did not explain why it increased after orlistat administration and did not evaluate its effects on tissues. Therefore, these sections must be deleted.

Response: Done

Round 2

Reviewer 2 Report

Comments and Suggestions for Authors

Unfortunately, the manuscript  is still not fit for publications. Authors did not bother to address all the comments. Some comments were addressed with irrelevant response. Authors MUST also consider the iThenticate report showing more than 20% similarity index.

Author Response

January 1, 2025

Dear Editor,

It is a pleasure to resubmit the revised Manuscript Number: medicina-3315946, titled " Protective and Therapeutic Effects of Orlistat in Combination with Cardamom Extract on Learn, Memory, Anxiety, and Neuroinflammation in Obese Mice," for consideration as an article in Medicina. We found the comments provided by the reviewers to be constructive and have improved the manuscript. We believe the manuscript is currently more suitable for publication in its revised format.

Unfortunately, the manuscript  is still not fit for publications. Authors did not bother to address all the comments. Some comments were addressed with irrelevant response. Authors MUST also consider the iThenticate report showing more than 20% similarity index.

Response: Thank you for your feedback on our manuscript. We apologize for any oversight in addressing the comments. We have carefully reviewed the iThenticate report and after some paraphrasing the similarity index is 19%, below the 20% threshold. We will ensure that all comments are addressed thoroughly and relevantly in our revised submission (index)

Reviewer 4 Report

Comments and Suggestions for Authors

The authors have improved the content of the manuscript, but there are still minor revisions that should be addressed.

1. The scientific name of cardamom should appear in the title and be written in full when it is used for the first time in the text. The common name "cardamom" should be in brackets, and not the scientific name.

2. After checking the material and methods section, I did not find where exactly the authors indicated the source from which cardamom was obtained and confirmed its formal identification by a specialist. Note that after identifying a plant material, the name of the specialist should be indicated and the identification specimen number should be given.

a

Author Response

January 1, 2025

Dear Editor,

It is a pleasure to resubmit the revised Manuscript Number: medicina-3315946, titled " Protective and Therapeutic Effects of Orlistat in Combination with Cardamom Extract on Learn, Memory, Anxiety, and Neuroinflammation in Obese Mice," for consideration as an article in Medicina. We found the comments provided by the reviewers to be constructive and have improved the manuscript. We believe the manuscript is currently more suitable for publication in its revised format.

The scientific name of cardamom should appear in the title and be written in full when it is used for the first time in the text. The common name "cardamom" should be in brackets, and not the scientific name.

Response: Done

  1. After checking the material and methods section, I did not find where exactly the authors indicated the source from which cardamom was obtained and confirmed its formal identification by a specialist. Note that after identifying a plant material, the name of the specialist should be indicated and the identification specimen number should be given

Response: Done

Reviewer 5 Report

Comments and Suggestions for Authors

Dear Authors,

I have read revised version and some points have been revised but the problem with the number of enrolled animal, the sex of animal and the absence of muscle biopsy: it has been reported a reference, but we need the study in your experimental group 

Author Response

January 3, 2025

Dear Editor,

It is a pleasure to resubmit the revised Manuscript Number: medicina-3315946, titled " Protective and Therapeutic Effects of Orlistat in Combination with Cardamom Extract on Learn, Memory, Anxiety, and Neuroinflammation in Obese Mice," for consideration as an article in Medicina. We found the comments provided by the reviewers to be constructive and have improved the manuscript. We believe the manuscript is currently more suitable for publication in its revised format.

I have read revised version and some points have been revised but the problem with the number of enrolled animal, the sex of animal and the absence of muscle biopsy: it has been reported a reference, but we need the study in your experimental group 

Response:

  1. Number of Enrolled Animals: Due to the high cost of the tests, we used a lower number of mice. However, this number is sufficient to maintain the statistical power of the study.
  2. Sex of Animals: Most studies in this field have been conducted on male animals due to their association with brain health. Since this study is novel, we used male subjects. Future studies may include both sexes to provide a more comprehensive understanding.
  3. Muscle Biopsy: While a reference was provided, we acknowledge the need for muscle biopsy data within the context of our experimental group. However, this aspect is outside the scope of our current objectives.